# Immune Phenotype as a Biomarker for Systemic Lupus Erythematosus

**DOI:** 10.3390/biom13060960

**Published:** 2023-06-08

**Authors:** Shingo Nakayamada, Yoshiya Tanaka

**Affiliations:** The First Department of Internal Medicine, School of Medicine, University of Occupational and Environmental Health, Japan, 1-1 Iseigaoka, Yahata-nishi-ku, Kitakyushu 807-8555, Fukuoka, Japan; s-nakaya@med.uoeh-u.ac.jp

**Keywords:** systemic lupus erythematosus, precision medicine, biomarkers, immunophenotyping

## Abstract

The treatment of rheumatoid arthritis was revolutionized with the use of molecular-targeted drugs that target immunoregulatory molecules. The success of treatment with these drugs prompted the development of molecular-targeted drugs for systemic lupus erythematosus. However, systemic lupus erythematosus is a disease with high heterogeneous immune abnormalities, and diverse cells or molecules can be treatment targets. Thus, the identification of subpopulations based on immune abnormalities is essential for the development of effective treatment. One analytical method used to identify subpopulations is the immunophenotyping of peripheral blood samples of patients. This analysis evaluates the validity of target molecules for peripheral blood immune cell subsets, which are expected to be developed as biomarkers for precision medicine in which appropriate treatment targets are set for each subpopulation.

## 1. Introduction

Systemic lupus erythematosus (SLE) is an autoimmune disease caused by the breakdown of immune tolerance that originates from genetic predisposition and environmental factors. SLE is characterized by the production of various autoantibodies and multiorgan damage [1,2]. For more than a half-century, the mainstay of SLE treatment was glucocorticoids (GCs) and immunosuppressants. These treatments contributed to the improvement of organ damage and acute-phase prognosis. However, treatment with these drugs is not specific to the pathology of SLE. The quality of life of patients with SLE decreases because of accumulation of organ damage caused by the disease and the treatment drugs, and the long-term prognosis is still insufficient. The goal of SLE treatment is remission without relapse or organ damage. To achieve this goal, it is necessary to minimize drug toxicity and avoid disease- and drug-associated organ damage [3]. Thus, the development of new molecular-targeted drugs that target immune abnormalities in SLE is expected.

Introduction of biological agents and Janus kinase (JAK) inhibitors that target molecules that are crucial in the pathogenesis of autoimmune diseases led to a paradigm shift in treatment [4]. In rheumatoid arthritis (RA), the treatment goal is clinical remission without structural damage or dysfunction of the joints. Molecular-targeted drugs used in RA treatment, which target tumor necrosis factor, interleukin 6 (IL-6), etc., were expanded to many other autoimmune diseases [4]. In contrast to these successes, the development of molecular-targeted drugs for SLE proved difficult [5]. Despite the approvals of the anti-B cell activating factor (BAFF) antibody belimumab and the anti-type I interferon (IFN) receptor antibody anifrolumab, most clinical trials of many molecular-targeted drugs failed [2]. One of the various possible causes of these failures is the pronounced heterogeneity of immune abnormalities in SLE. Diverse cells or molecules can be targeted for treatment. However, if the efficacy of molecular-targeted drugs is not consistent, it may not be possible to prove their efficacy in the patient population as a whole. For example, anifrolumab is significantly more effective in a group with higher expression of IFN-related genes, and classification of the patients into only two subpopulations (high or low IFN signature) allows the selection of effective drugs [6,7]. As SLE is heterogeneous, a patient population should ideally be selected that represents a homogeneous subset of SLE signatures. In autoimmune diseases, immunocompetent cells circulating in peripheral blood accurately reflect pathological conditions in tissues. Stratification of patients was previously attempted by performing flow cytometry to analyze immunophenotypes of immunocompetent cells. Here, we review the latest findings on immunophenotypes in SLE using data that were mainly obtained from our institution.

## 2. Immunopathogenesis of SLE

In the pathogenesis of SLE, a combination of genetic predisposition and epigenomic modification triggered by environmental factors causes the breakdown in autoimmune tolerance [1]. The pathology of SLE includes complex immune abnormalities associated with both acquired and innate immunity. Specifically, abnormalities of acquired immunity include excessive activation and abnormal differentiation of autoreactive T cells and enhanced differentiation of B cells to antibody-producing cells. The abnormalities of innate immunity include excessive production of type I IFN by dendritic cells (DCs) that recognize immune complexes consisting of self-nucleic acid and autoantibodies. The following sections briefly describe the immune abnormalities in SLE.

### 2.1. Bridge from Innate to Acquired Immunity

In SLE, autologous DNA and RNA released from damaged cells and neutrophils after the formation of neutrophil extracellular traps (NETosis) activate innate immunity through Toll-like receptor (TLR) 7/9. The activated innate immunocompetent cells activate phagocytic capacity and T cells through antigen presentation to T cells and stimulation via costimulators, such as CD80 and CD86, as well as various cytokines [8]. In particular, activation of plasmacytoid DCs (pDCs) leads to the secretion of a large amount of type I IFN [9]. Furthermore, type I IFN is involved in many immune abnormalities, such as maturation of monocytes to DCs, BAFF production from myeloid cells, differentiation of B cells to plasmocytes, autoantibody production, and induction of class switching [10,11,12,13,14].

### 2.2. Abnormal Differentiation and Activation of T Cells

Upon antigen presentation by DCs, T cells differentiate into functional subsets though stimulation by costimulatory molecules (e.g., CD 80/86) and various cytokines. Many recent reports suggested that the importance of T follicular helper (Tfh) cells in SLE [15,16,17,18,19]. Tfh cells are helper T cells that induce activation of B cells and antibody production in the germinal center [20]. Evidence from various studies showed that the levels of circulating Tfh-like cells, which are defined by the expression of CXC chemokine receptor 5 (CXCR5), inducible T cell costimulator (ICOS), programmed cell death 1 (PD-1), IL-21, and B cell lymphoma 6 (Bcl6), are increased in the peripheral blood of patients with SLE, and that the percentages of these cells correlate with disease activity, organ damage, and titers of anti-double-stranded DNA (dsDNA) antibodies and other autoantibodies [15,16,17,18,19]. T peripheral helper (Tph) cells recently attracted attention as a subset that exists in peripheral tissues and has a function similar to that of Tfh cells. A 2017 paper reported that CD4^+^CXCR5^−^PD-1^hi^ICOS^hi^ cells present in the synovial tissues of patients with RA produce IL-21 and CXC chemokine ligand (CXCL) 13, as well as induce the differentiation of plasmocytes and formation of ectopic lymph follicles [21]. Another study reported that CD4^+^CXCR5^−^CXCR3^+^PD-1^hi^IFN-γ^+^IL-10^+^ cells with a phenotype similar to that of Tph cells exist in the peripheral blood and lupus nephritic tissues of patients with SLE [22].

However, quantitative and qualitative abnormalities in regulatory T (Treg) cells were reported in SLE. The number and function of forkhead box protein P3 (FoxP3)-positive Treg cells are decreased in peripheral blood of patients with SLE [23]. T regulatory (Tfr) cells are a subset that inhibits Tfh cells [24,25,26,27]. Tfr cells are defined as CD4^+^CXCR5^+^ PD-1^+^ICOS^+^Bcl6^+^CD25^+^Foxp3^+^ cells. They produce IL-10, transforming growth factor-beta (TGF-β), granzyme, and other molecules, and inhibit B cell responses in the germinal center [27]. The percentage of Tfr cells in the peripheral blood of patients with SLE was conversely reported to be both decreased [28] and increased [29]. An analysis that included untreated patients showed that Tfr cells increased in association with decreased numbers of Tfh cells. Since the percentage of Tfr cells positively correlates with disease activity and autoantibody titers, this increase in Tfr cells may be a consequence of a secondary feedback mechanism associated with increased disease activity [29]. If Tfh and Tfr cells are plastic, the imbalance between these cells may be associated with the pathogenesis of SLE.

### 2.3. Abnormal Differentiation and Activation of B Cells

B cells acquire an affinity to autoantigens when they are stimulated through TLR7/9 by dsDNA and single-stranded RNA (ssRNA) or when immunoglobulin (Ig) genes are reconstituted. In this process, stimulation by IL-21 and IFN-γ produced by Tfh cells, in addition to antigen stimulation through B cell receptors, induces differentiation to autoreactive B cells and production of autoantibodies [30]. Furthermore, CD40, CD80/86, and inducible costimulator ligand (ICOSL) are expressed on the surface of B cells, and CD40L, CD28, and ICOS are expressed on the surface of Tfh cells. Tfh and B cells interact through costimulatory molecules and mutually activate each other [31]. BAFF, which is produced by monocytes, macrophages, DCs, T cells, and other cells, binds to three BAFF receptors (BAFFRs), transmembrane activator and calcium modulating ligand interactor protein (TACI), and B cell maturation antigen (BCMA) on B cells. The binding induces B cells to survive, proliferate, switch classes, and differentiate into plasmocytes [32,33]. A proliferation-inducing ligand (APRIL) binds to BCMA, TACI, and proteoglycan to stimulate B cells and promote B cell activation [32]. BAFF and APRIL levels reportedly correlate with SLE activity [34].

The B cell population characterized by CD11c^+^T-box-expressed-in-T (T-bet)^+^ cells recently attracted scientific attention. In 2010, Mellor et al. reported a subset of B cells in mice that coexpress the DC marker CD11c, immunoglobulins (IgM, Igκ, and IgD), indoleamine2,3-dioxygenase (IDO), and the B cell transcription factor PAX5 [35]. Another study demonstrated that this subset of B cells emerges with aging in normal mice and proliferates in various autoimmune diseases. These cells were termed age- and/or autoimmunity-associated B cells (ABCs) [36]. Differentiation of ABCs is promoted via the stimulation of naïve B cells through TLR7, IFN-γ, IL-21, and other molecules and depends on the transcription factor T-bet. ABCs differentiate into antibody-producing cells. In a mouse model of lupus, ABCs were reported to appear at early stages and produce pathogenic autoantibodies [37].

Human B cells also contain ABC-like cells. These cells are termed activated naïve B cells (CD19^hi^, IgD^+^, CD27^−^, CD24^−^, MTG^+^, CD38^−^, C11c^+^, and T-bet^+^) or double negative (DN) 2 cells (CD19^+^, IgD^−^, CD27^−^, CXCR5^−^, C11c^+^, and T-bet^+^). The number of these cells increases in the peripheral blood and diseased tissues of patients with SLE. These activated cells are derived from naïve B cells and directly differentiate into antibody-producing cells. This approach differs from the conventional pathway of differentiation of class-switched memory B cells to antibody-producing cells. Compared to other B cell subsets, DN2 cells were reported to increase in patients with renal lesions; correlate with disease activity, anti-Sm antibody, and anti-ribonucleoprotein antibody; and express a higher level of interferon regulatory factor 4, which is a marker of plasmocyte differentiation [38]. DN2 cells appear to be involved in the pathology of SLE.

## 3. Immunophenotyping via Multicolor Flow Cytometry

Immune cells function within lymphoid and peripheral tissues and play a role in pathogenesis. Thus, tissue analysis can directly elucidate the pathology. However, it is difficult to perform multiple tissue biopsies in patients. Immunocompetent cells circulating in peripheral blood accurately reflect pathological conditions in tissues. Analysis of cell surface antigens on peripheral blood lymphocytes allows the subclassification of pathological conditions and patients based on differences in the expression of these antigens. Flow cytometry is widely used and was intensively evaluated over the past 40 years as a useful analytical tool that can detect cell-surface antigens on immunocompetent cells [39,40]. In flow cytometry, molecules expressed on cells are stained with specific fluorescent-labeled antibodies. The cells are carried via a laminar flow of liquid past a beam of laser light to optically detect the molecular expression levels of individual cells. In 2012, the Human Immunology Project Consortium, which is mainly funded by the National Institutes of Health, proposed a standardized protocol for phenotyping of human immune cells using multicolor flow cytometry [41]. We prepared an antibody cocktail for each target cell, including T cells, B cells, natural killer cells, monocytes, and DCs, by combining eight types of antibodies with different surface antigens of immune cells in peripheral blood and eight colors of fluorescence (Figure 1) [42,43]. Multicolor analysis using these antibody cocktails allows simultaneous detection of phenotypes of multiple different cells and different subsets. In addition, interactions between cell populations can also be analyzed under conditions that are similar to in vivo conditions. In particular, useful information applicable to clinical practice can be obtained by analyzing the subsets according to the differences in the expression of chemokine receptors; expression of CD45RA/RO, IgD/CD27, etc.; and activation according to the expression of CD69, ICOS, CX3CR1, and others.

## 4. Immune Phenotype in Patients with SLE

We attempted to understand the pathology and evaluate the validity of molecular targets by collecting peripheral blood samples from healthy individuals and patients with autoimmune diseases and performing multicolor flow cytometry to analyze immune phenotypes, as described above [42,43]. We reported peripheral blood immunophenotyping performed in 143 patients with SLE [44]. The percentages of Treg and Tfh cells in peripheral blood were higher in patients with SLE than in healthy individuals, and enhanced plasmocyte differentiation correlated with disease activity. In the cluster analysis, patients with SLE were classified into three groups according to differences in T cell phenotypes, in addition to common abnormalities of B cell differentiation. The first group displayed poor differentiation and less abnormal activation of T cells (T cell-independent group). The second group displayed marked increases in memory Treg cells (Treg-dominant group). The third group displayed marked increases in Tfh cells and plasmocytes (Tfh-dominant group). Patients with active SLE were divided into three subpopulations according to differences in T cell phenotypes. Among these subpopulations, the Tfh dominant group of patients was resistant to existing immunosuppressive therapies. Furthermore, we reported the enhanced activation of Tfh/T helper 1 (Th1)-like (Tfh1) cells, which exhibit phenotypes of both Tfh and Th1 cells, in the peripheral blood of patients with SLE [45]. The induction of Tfh cells was determined via epigenomic regulation through histone protein modification during the activation of signal transducer and transcription STAT1 and STAT4 through IL-12. These results suggest that Tfh cells might be a new target for the treatment of SLE (Figure 2) [46].

The plasticity of Tfh and Tfr cells is also a recent research topic. The percentage of CD4^+^CXCR5^+^Foxp3^−^PD-1^hi^ Tfh cells was reportedly higher in the peripheral blood of patients with active SLE than in the peripheral blood of healthy individuals, whereas the percentage of CD4^+^CXCR5^+^CD45RA^−^Foxp3^hi^-activated Tfr cells was lower [47]. Moreover, serum IL-2 concentrations were lower in patients with SLE than in healthy individuals. In vitro, memory Tfh cells extracted from peripheral blood and stimulated with IL-2 were differentiated and converted to functional Tfr cells (CXCR5^+^Bcl6^+^Foxp3^hi^ cells). The differentiation and conversion occurred when STAT3 and STAT5 activated via IL-2 directly bound to the FOXP3 and BCL6 gene loci and inhibited H3K27me3, which is an inhibitory histone marker [47,48]. These results suggest that the balance between Tfh and Tfr cells is skewed to Tfh cells in patients with active SLE, which may depend on the deficiency of IL-2 in patients with SLE [48]. Interestingly, the gene loci of IL-2 and IL-21 are located in close proximity. We reported the association between genomic abnormalities in these regulatory regions and Tfr cells [49]. The single nucleotide polymorphism rs62324212 (C>A) in the IL 21 antisense RNA (IL21-AS1) is a genetic risk variant associated with SLE. Analysis using the Ensembl Genome Browser revealed that rs62324212 is located in the predicted enhancer region of IL21-AS1. IL21-AS1 was reportedly expressed in the nuclei of CD4^+^ T and B cells, though its expression was decreased in patients with SLE. Importantly, the expression of IL21-AS1 was positively correlated with the expression of IL-2 at the messenger RNA levels rather than with the expression of IL-21 and was associated with the percentage of activated Tfr cells. In addition, a significantly negative correlation was observed between the expression of IL21-AS1 and disease activity in patients with SLE. Based on these collective findings, IL21-AS1 appears to be involved in the activation of Tfr cells through IL-2 and affects disease activity in SLE [49].

Recent studies reported that the number of Tph cells is increased in the peripheral blood and nephritic tissues of patients with SLE [22]. The mechanism for the induced differentiation of Tph cells remains unknown. The differentiation of Tph cells was recently reported to be dependent on type I IFN [50]. In addition, we found that the differentiation of PD-1^hi^MAF (macrophage activating factor)^+^IL-21^+^IL-10^+^ Tph-like cells, which are considered important in SLE, can be induced by TGF-β3 [51]. Tph-like cells induced by TGF-β3 efficiently promoted the differentiation of class-switched memory B cells to plasmocytes and enhanced antibody production. Importantly, TGF-β3 was highly expressed in macrophages in renal tissues of patients with active lupus nephritis; the elevated expression correlated with the accumulation of CD4^+^CXCR5^−^PD-1^+^ Tph cells in renal tissues. These results suggest that the induction of Tph cells by TGF-β3 plays an important role in the pathogenesis of lupus nephritis.

## 5. Challenges to Precision Medicine 

In his 2015 State of Union address, President Barack Obama launched the Precision Medicine Initiative. The concept of precision medicine is that treatment and preventive strategies are established for each subpopulation of cancer patients based on the analytical studies of the enormous biological data available, such as genome data. Treatment according to subpopulations was also expected to be realistic in terms of practice and costs [52,53]. Such precision medicine is only achievable using molecular-targeted therapy and should be applicable to SLE, which is highly heterogeneous. The implementation of precision medicine requires complex biomolecular information that combines clinical, genomic, and immunity information, such as cell surface antigens and cytokines, as well as other information. However, this scope of information is still lacking for autoimmune diseases. To provide genomic information, a large-scale, multi-center project using single-cell analysis of tissues and other tools (Accelerating Medicines Partnership) is being conducted, mainly in the United States.

In 2016, Banchereau et al. published the transcriptome analysis results for peripheral blood; this analysis was performed as part of a study that followed 158 pediatric patients with SLE for 4 years [54]. As with our studies, the authors reported that in addition to the presence of the common IFN signature, the plasmocyte signature most strongly reflected disease activity. Importantly, patients with SLE were stratified into seven subpopulations based on differences in immune-related single nucleotide polymorphisms and disease activity. The authors also revealed the heterogeneity of SLE at the molecular level. These results may also indicate one of the factors for the failure of clinical trials of molecular-targeted drugs for SLE. It may be difficult to prove the effects of molecular-targeted drugs when patients with SLE are treated as a clinically homogeneous population.

As shown in our aforementioned study, in subpopulations of patients with SLE, T cells exhibit enhanced IL-12/tyrosine kinase (TYK) 2/STAT4 signaling, and Tfh/Th1-like cells are activated [45]. Reports from other institutions showed that in patients with risk alleles of STAT4, phosphorylation of STAT4 in T cells is enhanced via stimulation with IL-12 or type I IFN and inhibited using JAK2 inhibitors and TYK2 inhibitors [55]. This result suggests that anti-IL-12/23 antibodies and JAK inhibitors are beneficial for patients in whom Tfh cells are dominantly activated or patients with risk alleles of STAT4. In addition, another study evaluated the function of the STAT1 and STAT4 loci. Patel et al. reported a plausible mechanism of increased lupus risk at the STAT1–STAT4 locus, in which the rs11889341 risk allele caused elevated STAT1 expression in B cells through reduced repressor activity via increased binding of HMGA1 [56]. 

Recent emerging studies also demonstrated the clinical diversity of SLE from cytokine, transcriptome, and genomic analyses. Lanata et al. identified three distinct clinical subtypes of SLE, which have distinct patterns of methylation at specific CpG sites, reflecting the influence of both genetic and non-genetic effects [57]. Oke et al. reported that highly functional type I IFN activity captures active SLE in most domains, though a more distinct pattern of organ involvement is associated with the profile of circulating IFNs, including type I, type II, and type III IFN [58]. Bradley et al. performed an unbiased transcriptome analysis. They found transcripts of hundreds of genes that were consistently altered in SLE T cell samples, highlighting the induction of pathways related to mitochondria, nucleotide metabolism, and DNA replication. Furthermore, T cell gene expression indicated the presence of several patient subtypes, such as having only a minimal expression phenotype, male type, or severe type with or without induction of genes associated with membrane protein production [59]. These findings could be useful in the development of biomarkers for patient stratification.

A collaborative study group consisting of the University of Tokyo and RIKEN published the results of the largest analysis performed to date. In this analysis, a cell sorter was used to sort 6386 samples of 27 types of immune cells collected from 136 patients with SLE and 89 healthy individuals who were included in the ImmuNexUT functional genome database. In the study, the gene expression levels were comprehensively assessed using RNA sequencing [60]. The authors described the involvement of different immune cells in the onset (disease-state signature) and exacerbation (disease-activity signature) of SLE. Symptoms were most strongly associated with the gene expression of Th1 cells in patients with skin symptoms, monocyte–lineage cells in patients with joint symptoms, and neutrophil–lineage cells in patients with renal symptoms. The findings suggest that different immune cells are activated for each symptom. Regarding the association with responses to therapeutic drugs, the differentially expressed genes identified before and after belimumab therapy accumulated in B cell lineage cells, while differentially expressed genes in patients receiving and not receiving oral mycophenolate mofetil accumulated in Th1 cells, memory CD8-positive T cells, and plasmocytes [60]. Furthermore, gene clusters suppressed by these therapeutic drugs extensively overlapped in the disease activity signature gene clusters and treatment-responsive patients [60]. The findings confirm that the current therapeutic drugs for SLE exert clinical effects by suppressing the disease activity signatures. These signatures may also be useful for exploring new therapeutic targets for SLE.

## 6. Conclusions

In recent years, immunophenotyping of SLE revealed extensively diverse immune abnormalities in SLE. Many disease susceptibility genes for SLE are associated with dendritic and lymphoid cell signaling. Thus, cells bridging the innate and adaptive immune systems, as well as signaling molecules from these cells, are promising therapeutic targets. Presently, several JAK inhibitors, the type II anti-CD20 antibody obinutuzumab, proteasome inhibitors (e.g., iberdomide), pDC-targeting drugs, and other treatments are in the development stage. Clinical trials are underway to evaluate drugs targeting Tfh cells, ABCs, and other targets, such as rozibafusp alfa (bispecific monoclonal antibody to ICOSL and BAFF) and prezalumab (dual antagonist to ICOS and CD28). However, for the treatment of SLE, which is highly heterogeneous, it is essential to establish new treatment systems and treatment strategies that enable the differential use of molecular-targeted drugs according to pathology. There is great expectation for this precision medicine. The implementation of precision medicine requires classification of patients into subpopulations based on the analysis of complex biomolecular information that combines immunophenotypes, genome, transcriptome, and other aspects with the selection of appropriate molecular-targeted drugs for each subpopulation. Bidirectional research translation between the bench and the bedside is expected to lead to the elucidation of SLE pathology and advances in the treatment of SLE.

## Figures and Tables

**Figure 1 biomolecules-13-00960-f001:**
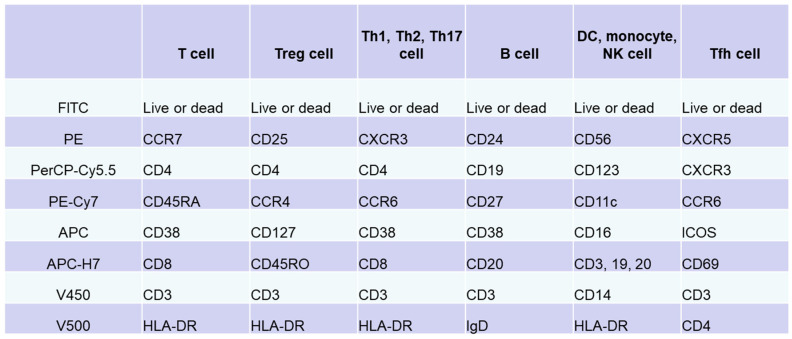
Immunophenotyping via flow cytometry analysis of peripheral blood lymphocytes. Based on standardized protocol for phenotyping of human immune cells using multicolor flow cytometry [41,42,43], we adjusted an antibody cocktail for each target cell (T cells, B cells, natural killer cells, monocytes, or dendritic cells) by combining eight types of antibodies against surface antigens of immune cells in peripheral blood and eight colors. In addition to these antibody cocktails, original set for evaluation of Tfh cells was prepared.

**Figure 2 biomolecules-13-00960-f002:**
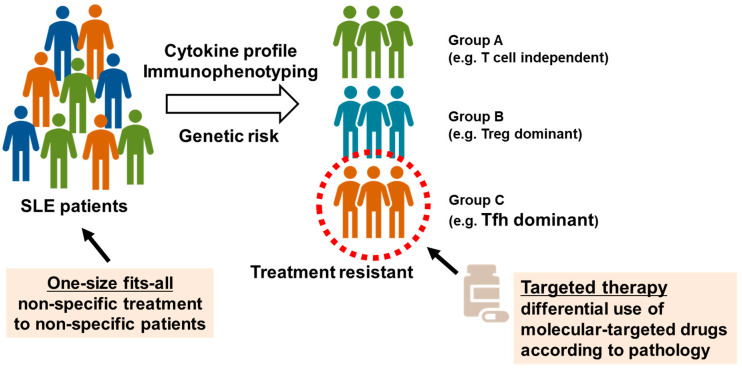
Identification of systemic lupus erythematosus (SLE) subpopulations via peripheral blood immunophenotyping. Based on peripheral blood immunophenotyping of 143 patients with SLE, patients with highly active SLE were classified into three subpopulations: poor differentiation and less abnormal activation of T cells (T cell independent group), marked increases in memory regulatory T (Treg) cells (Treg dominant group), and marked increases in T follicular helper (Tfh) cells and plasma cells (Tfh dominant group). Tfh dominant group is resistant to existing immunosuppressants, and new treatment strategies may be needed.

## Data Availability

No new data were created.

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
