# Peer review of "Immune Phenotype as a Biomarker for Systemic Lupus Erythematosus"

_biomolecules, 2023, doi:10.3390/biom13060960_

Round 1

Reviewer 1 Report

This review processes a current paper about Immune Phenotype as a Biomarker for Systemic Lupus Erythematosus. The development of new biomarkers and molecular-targeted drugs is a priority in SLE as well as in other autoimmune diseases, based on the heterogeneity of autoimmune processes. The structure of the article is well-built and logical. The figure and table are clear, as is the related explanation. The introduction and the immunopathogenesis of the SLE chapter are well structured and appropriate in terms of content.

The study’s major limitation is that the references are incomplete. A review of relevant references is necessary, especially of the most recent publications

Minor issues to be addressed are:

1.           Line 102 The abbreviation Tf needs to be explained.

2.           Line 178 The abbreviation XX needs to be explained.

Author Response

A point-by-point response to the comments of Reviewer #1

Comments to the Author

This review processes a current paper about Immune Phenotype as a Biomarker for Systemic Lupus Erythematosus. The development of new biomarkers and molecular-targeted drugs is a priority in SLE as well as in other autoimmune diseases, based on the heterogeneity of autoimmune processes. The structure of the article is well-built and logical. The figure and table are clear, as is the related explanation. The introduction and the immunopathogenesis of the SLE chapter are well structured and appropriate in terms of content.

Response: We sincerely thank the reviewer for the constructive comments and suggestions, which helped us to substantially improve our manuscript.

The study’s major limitation is that the references are incomplete. A review of relevant references is necessary, especially of the most recent publications.

Response: We appreciate the reviewer’s suggestion. Reviewer #2 also made the same point. We have added the following references.

  1. Gravallese EM, Firestein GS. Rheumatoid Arthritis - Common Origins, Divergent Mechanisms. N Engl J Med. 2023, 388, 529-542.
  2. Murphy G, Isenberg DA. New therapies for systemic lupus erythematosus - past imperfect, future tense. Rev. Rheumatol. 2019, 15, 403-412.
  3. Simpson N, Gatenby PA, Wilson A, et al: Expansion of circulating T cells resembling follicular helper T cells is a fixed phenotype that identifies a subset of severe systemic lupus erythematosus. Arthritis Rheum 2010, 62, 234-244.
  4. Wong CK, Wong PT, Tam LS, et al: Elevated production of B cell chemokine CXCL13 is correlated with systemic lupus erythematosus disease activity. J Clin Immunol 2010, 30, 45-52.
  5. Choi JY, Ho JH, Pasoto SG, et al: Circulating follicular helper-like T cells in systemic lupus erythematosus: association with disease activity. Arthritis Rheumatol 2015, 67, 988-999.
  6. Arazi A, Rao DA, Berthier CC, et al: The immune cell landscape in kidneys of patients with lupus nephritis. Nat Immunol 2019, 20, 902-914.
  7. Perez OD, et al. Multiparameter analysis of intracellular phosphoepitopes in immunophenotyped cell populations by flow cytometry. Curr Protoc Cytom. 2005, 32, 6.20.1–6.20.22.
  8. Parish CR, et al. Use of the intracellular fluorescent dye CFSE to monitor lymphocyte migration and proliferation. Curr Protoc Immunol. 2009, 84, 4.9.1–4.9.13.
  9. Patel ZH, et al. A plausibly causal functional lupus-associated risk variant in the STAT1–STAT4 locus. Hum Mol Genet. 2018, 27, 2392-2404.
  10. Lanata CM, et al. A phenotypic and genomics approach in a multi-ethnic cohort to subtype systemic lupus erythematosus. Nat Commun. 2019, 10, 3902.
  11. Oke V, et al. High levels of circulating interferons type I, type II and type III associate with distinct clinical features of active systemic lupus erythematosus. Arthritis Res Ther. 2019, 21, 107.
  12. Bradley SJ, et al. T Cell Transcriptomes Describe Patient Subtypes in Systemic Lupus Erythematosus. PLoS One. 2015, 10, e0141171.

Minor issues to be addressed are:

  • Line 102 The abbreviation Tf needs to be explained.

Response: We apologize for the mistake. The "f" in Tf is unnecessary and has been removed.

  • Line 178 The abbreviation XX needs to be explained.

Response: We apologize for the mistake. The “XX” has been revised to “41, 42” (Maecker HT, et al. Nat Rev Immunol. 2012; Nakayamada S, et al. Rheumatology (Oxford) 2018).

Reviewer 2 Report

In the review article, Immune Phenotype as a Biomarker for Systemic Lupus Erythematosus, the authors describe the heterogenous immune features of SLE patients and how identification of these cell subpopulations could provide opportunities for precision medicine initiatives. 

Effective treatment strategies for SLE remains a challenge and this review article covers an important topic that is of continual interest to the research community. However, there are several concerns with the current manuscript which are listed below. 

Main Concerns:

-The review’s references appear largely skewed towards work by the authors. While the authors have clearly contributed to this topic and the field, a review should include a broader range of studies by other groups. Furthermore, the figures included in this review appear to be a summary of the author’s previously published work. There needs to be more integration of findings from other studies. In another example, line 85 lists “many recent reports have suggested…” but only one reference is listed, which is a review by the current authors. Please include more primary references in this manuscript. Doing so will further its utility to readers.  

There are areas lacking citations in the manuscript. Some examples:

--Lines 39-45, no references regarding biological agents for SLE or RA treatments. 

--Missing reference (currently as XX) on line 178 in Figure 1 caption.

--Need citations for flow cytometry overview (starting section 3. Line 151).

--Starting line 289: unclear if these findings are related to reference 48 or a different study (no citation).

While a review article can’t list every relevant paper, there seems to be a number of relevant citations missing. Some quick examples that stand-out (not an exhaustive list) 

--line 272: there are other studies that have evaluated functionality of the STAT4 locus, e.g.: Patel et al. A plausibly causal functional lupus-associated risk variant in the STAT1–STAT4 locus

--Lanata et al. A phenotypic and genomics approach in a multi-ethnic cohort to subtype systemic lupus erythematosus

--Oke et al. High levels of circulating interferons type I, type II and type III associate with distinct clinical features of active systemic lupus erythematosus

--Bradley et al. T Cell Transcriptomes Describe Patient Subtypes in Systemic Lupus Erythematosus.

Minor Suggestions:

 Several places in the manuscript would benefit from additional editing or clarity. Some examples:

--line 35 reads as repetitive. Does the second “organ damage” refer to drug-induced organ damage or SLE-induced organ damage?  “The goal of SLE treatment is remission  without  relapse  or  organ  damage.  To  achieve  this  goal,  it  is  necessary  to  minimize drug  toxicity  and  avoid  organ  damage.” 

--lines 54-56 are unclear. SLE is heterogenous, so ideally patient populations should be selected for homogenous subsets of SLE signatures, correct?

In conclusion, the authors have presented an important topic in this review article. They highlight key challenges in developing effective SLE treatments (e.g., line 270 “It  may  be  difficult  to  prove  the  effects  of  molecular-targeted  drugs  when  patients with SLE are treated as a clinically homogeneous population.”)

However, key concerns for the manuscript center on a limited set of citations and would suggest the authors include and summarize findings from additional relevant works. 

Overall, no major concerns on writing. 

Author Response

A point-by-point response to the comments of Reviewer #2

Comments to the Author

In the review article, Immune Phenotype as a Biomarker for Systemic Lupus Erythematosus, the authors describe the heterogenous immune features of SLE patients and how identification of these cell subpopulations could provide opportunities for precision medicine initiatives.

Effective treatment strategies for SLE remains a challenge and this review article covers an important topic that is of continual interest to the research community. However, there are several concerns with the current manuscript which are listed below.

Response: We sincerely thank the reviewer for the constructive comments and suggestions, which helped us to improve our manuscript substantially.

Main Concerns:

The review’s references appear largely skewed towards work by the authors. While the authors have clearly contributed to this topic and the field, a review should include a broader range of studies by other groups. Furthermore, the figures included in this review appear to be a summary of the author’s previously published work. There needs to be more integration of findings from other studies. In another example, line 85 lists “many recent reports have suggested…” but only one reference is listed, which is a review by the current authors. Please include more primary references in this manuscript. Doing so will further its utility to readers.

Response: We appreciate the reviewer’s suggestion. Reviewer #1 also made the same point. We have added the following reference in the relevant parts.

  1. Simpson N, Gatenby PA, Wilson A, et al: Expansion of circulating T cells resembling follicular helper T cells is a fixed phenotype that identifies a subset of severe systemic lupus erythematosus. Arthritis Rheum 2010, 62, 234-244.
  2. Wong CK, Wong PT, Tam LS, et al: Elevated production of B cell chemokine CXCL13 is correlated with systemic lupus erythematosus disease activity. J Clin Immunol 2010, 30, 45-52.
  3. Choi JY, Ho JH, Pasoto SG, et al: Circulating follicular helper-like T cells in systemic lupus erythematosus: association with disease activity. Arthritis Rheumatol 2015, 67, 988-999.
  4. Arazi A, Rao DA, Berthier CC, et al: The immune cell landscape in kidneys of patients with lupus nephritis. Nat Immunol 2019, 20, 902-914.

There are areas lacking citations in the manuscript. Some examples:

--Lines 39-45, no references regarding biological agents for SLE or RA treatments.

Response: We have added the following references.

  1. Gravallese EM, Firestein GS. Rheumatoid Arthritis - Common Origins, Divergent Mechanisms. N Engl J Med. 2023, 388, 529-542.
  2. Murphy G, Isenberg DA. New therapies for systemic lupus erythematosus - past imperfect, future tense. Rev. Rheumatol. 2019, 15, 403-412.

--Missing reference (currently as XX) on line 178 in Figure 1 caption.

Response: We apologize for the mistake. The “XX” has been revised to “41, 42” (Maecker HT, et al. Nat Rev Immunol. 2012; Nakayamada S, et al. Rheumatology (Oxford) 2018).

--Need citations for flow cytometry overview (starting section 3. Line 151).

Response: We have added the following references.

  1. Perez OD, et al. Multiparameter analysis of intracellular phosphoepitopes in immunophenotyped cell populations by flow cytometry. Curr Protoc Cytom. 2005, 32, 6.20.1–6.20.22.
  2. Parish CR, et al. Use of the intracellular fluorescent dye CFSE to monitor lymphocyte migration and proliferation. Curr Protoc Immunol. 2009, 84, 4.9.1–4.9.13.

--Starting line 289: unclear if these findings are related to reference 48 or a different study (no citation).

Response: Since all of these are information from the same paper (#60), we have specified the citation.

While a review article can’t list every relevant paper, there seems to be a number of relevant citations missing. Some quick examples that stand-out (not an exhaustive list)

--line 272: there are other studies that have evaluated functionality of the STAT4 locus, e.g.: Patel et al. A plausibly causal functional lupus-associated risk variant in the STAT1–STAT4 locus

--Lanata et al. A phenotypic and genomics approach in a multi-ethnic cohort to subtype systemic lupus erythematosus

--Oke et al. High levels of circulating interferons type I, type II, and type III are associate with distinct clinical features of active systemic lupus erythematosus

--Bradley et al. T Cell Transcriptomes Describe Patient Subtypes in Systemic Lupus Erythematosus.

Response: We have added the following sentence and references in the relevant sentence, as the reviewer kindly proposed. “Furthermore, there are other studies that have evaluated functionality of the STAT4 locus.56-59)

  1. Patel ZH, et al. A plausibly causal functional lupus-associated risk variant in the STAT1–STAT4 locus. Hum Mol Genet. 2018, 27, 2392-2404.
  2. Lanata CM, et al. A phenotypic and genomics approach in a multi-ethnic cohort to subtype systemic lupus erythematosus. Nat Commun. 2019, 10, 3902.
  3. Oke V, et al. High levels of circulating interferons type I, type II, and type III associate with distinct clinical features of active systemic lupus erythematosus. Arthritis Res Ther. 2019, 21, 107.
  4. Bradley SJ, et al. T Cell Transcriptomes Describe Patient Subtypes in Systemic Lupus Erythematosus. PLoS One. 2015, 10, e0141171.

Minor Suggestions:

 Several places in the manuscript would benefit from additional editing or clarity. Some examples:

--line 35 reads as repetitive. Does the second “organ damage” refer to drug-induced organ damage or SLE-induced organ damage?  “The goal of SLE treatment is remission  without  relapse  or  organ  damage.  To  achieve  this  goal,  it  is  necessary  to  minimize drug  toxicity  and  avoid  organ  damage.”

Response: We have revised the following sentences, “The goal of SLE treatment is remission without relapse or organ damage. To achieve this goal, it is necessary to minimize drug toxicity and avoid disease- and drug-associated organ damage.”

--lines 54-56 are unclear. SLE is heterogenous, so ideally patient populations should be selected for homogenous subsets of SLE signatures, correct?

Response: We have revised the text as the following, “Because SLE is heterogeneous, a patient population should ideally be selected that represents a homogeneous subset of SLE signatures.”.

In conclusion, the authors have presented an important topic in this review article. They highlight key challenges in developing effective SLE treatments (e.g., line 270 “It may be difficult to prove the effects of molecular-targeted drugs when patients with SLE are treated as a clinically homogeneous population.”)

However, key concerns for the manuscript center on a limited set of citations and would suggest the authors include and summarize findings from additional relevant works.

Response: Again, we sincerely thank the reviewer for the constructive comments and suggestions, which helped us to improve our manuscript substantially.

Round 2

Reviewer 1 Report

Dear Authors, 

Thank you very much for accepting the reviewer's proposal and for making the repairs. 

Yours sincerely

Author Response

Thank you so much. We were pleased to know of the acceptance of our manuscript for publication in Biomolecules.

Reviewer 2 Report

The authors have addressed several concerns since the first submission. Previously flagged sentences (e.g., lines 35-37) have been edited for clarity. Additional references have been added to the manuscript; however, the results of these references were neither discussed nor integrated within the text. 

Thus, I still have concerns that this review is not a general review of the subject and instead largely focuses on authors' and authors' institution's own work. However, I note that the authors acknowledge this in the introduction. This is not a format I would expect of a review article, but will leave it to the editors to assess if this is appropriate/consistent with this journal's review article guidelines. 

No concerns with quality of English language. 

Author Response

We sincerely thank the reviewer for the comments. According to the suggestion, we have added the following discussion and references.

“In addition, another study evaluated the function of the STAT1 and STAT4 loci. Patel et al. reported a plausible mechanism of increased lupus risk at the STAT1–STAT4 locus, in which the rs11889341 risk allele causes elevated STAT1 expression in B cells through reduced repressor activity by increased binding of HMGA1.56)

Recent emerging studies have also demonstrated the clinical diversity of SLE from cytokine, transcriptome, and genomic analyses. Lanata et al. identified three distinct clinical subtypes of SLE, which have distinct patterns of methylation at specific CpG sites, reflecting the influence of both genetic and non-genetic effects.57) Oke et al. reported that highly functional type I IFN activity captures active SLE in most domains, but a more distinct pattern of organ involvement is associated with the profile of circulating IFNs including type I, type II, and type III IFN.58) Bradley et al. performed an unbiased transcriptome analysis. They found transcripts of hundreds of genes that were consistently altered in SLE T cell samples, highlighting the induction of pathways related to mitochondria, nucleotide metabolism, and DNA replication. Furthermore, T cell gene expression indicated the presence of several patient subtypes, such as having only a minimal expression phenotype, male type, or severe with or without induction of genes associated with membrane protein production.59) These findings could be useful in the development of biomarkers for patient stratification.” (lines 292-310).

  1. Patel ZH, et al. A plausibly causal functional lupus-associated risk variant in the STAT1–STAT4 locus. Hum Mol Genet. 2018, 27, 2392-2404.
  2. Lanata CM, et al. A phenotypic and genomics approach in a multi-ethnic cohort to subtype systemic lupus erythematosus. Nat Commun. 2019, 10, 3902.
  3. Oke V, et al. High levels of circulating interferons type I, type II, and type III associate with distinct clinical features of active systemic lupus erythematosus. Arthritis Res Ther. 2019, 21, 107.
  4. Bradley SJ, et al. T Cell Transcriptomes Describe Patient Subtypes in Systemic Lupus Erythematosus. PLoS One. 2015, 10, e0141171.